# New Aspects Concerning the Ampicillin Photodegradation

**DOI:** 10.3390/ph15040415

**Published:** 2022-03-29

**Authors:** Radu Cercel, Mirela Paraschiv, Cristina Stefania Florica, Monica Daescu, Adelina Udrescu, Romeo C. Ciobanu, Cristina Schreiner, Mihaela Baibarac

**Affiliations:** 1National Institute of Materials Physics, Atomistilor Street 405A, P.O. Box MG-7, 077125 Bucharest, Romania; radu.cercel@infim.ro (R.C.); mirela.cristea@infim.ro (M.P.); stefania.florica@infim.ro (C.S.F.); monica.daescu@infim.ro (M.D.); adelina.matea@infim.ro (A.U.); 2SC All Green SRL, 8 George Cosbuc Str., 700470 Iasi, Romania; rciobanu@yahoo.com (R.C.C.); cschrein@ee.tuiasi.ro (C.S.); 3Faculty of Electrical Engineering, Department of Electrical Measurements and Materials, Technical University Gh. Asachi Iasi, Bd. Profesor Dimitrie Mangeron 67, 700050 Iasi, Romania

**Keywords:** ampicillin, photoluminescence, UV-VIS spectroscopy, photodegradation

## Abstract

New aspects concerning the photodegradation (PD) of ampicillin are reported by photoluminescence (PL), Raman scattering and FTIR spectroscopy. The exposure of ampicillin in the absence (AM) and in the presence of the excipient (AMP) to UV light leads to an intensity diminution of the photoluminescence excitation (PLE) and photoluminescence (PL) spectra and the emergence of a new IR band at 3450 cm^−1^. The photoluminescence studies demonstrate that the AM PD is amplified in the presence of excipients and an alkaline medium. In this last case, the PD process of AM involves the emergence of new compounds, whose presence is highlighted by: (i) the emergence of the isosbestic point at 300 nm in the UV-VIS spectra; (ii) a change in the ratio between the absorbance of IR bands situated in the spectral ranges 1200–1660 and 3250–3450 cm^−1^; and (iii) a change in the ratio between the intensities of the Raman lines localized in the spectral ranges 1050–1800 and 2750–3100 cm^−1^. A chemical mechanism of the PD processes of AM in an alkaline medium is proposed.

## 1. Introduction

Ampicillin is one of the earliest antibiotics discovered in 1958 [1] and is most commonly used to treat various diseases such as meningitis [2], respiratory tract infections [3], endocarditis [4], gynecological surgical procedures [5], cholera [6] and typhoid fever [7]. Recently, taking into account the therapeutic effect of ampicillin, devices were developed for bacterial detection based on surface enhancer Raman scattering and antimicrobial susceptibility testing [8]. Despite the therapeutic effect, the presence of ampicillin residue in the food products has induced an intense effort in order to diminution the risks concerning the population health [9], their uncontrolled assimilation inducing resistance to the administration of antibiotics during the treatment of various infections [10]. Various methods were used in order to characterize or detect the ampicillin, such as UV-VIS spectroscopy [11], IR spectroscopy [12], Raman scattering [12], X-ray diffraction (XRD) [13], and so on. Concerning the UV photolysis of ampicillin, this was studied by UV-VIS spectroscopy [14]. In comparison with this progress, this work reports new results obtained by photoluminescence (PL) and FTIR spectroscopy concerning the photodegradation (PD) of AM and AMP.

Photocatalytic degradation of ampicillin was studied in the presence of various compounds such as Fe_3_O_4_ nanoparticles decorated with Au [15], magnetite-metal organic framework [16], TiO_2_ [17], polylactic acid/TiO_2_nanofibers [18], and so on. In contrast with this progress, this work reports the influence of the alkaline media on ampicillin PD. In this order, the correlated studies of photoluminescence, UV-VIS spectroscopy and Raman scattering are shown.

## 2. Results

### 2.1. Optical and Structural Properties of AM and Its Photodegradation in Solid State

According to Figure 1 and PDF-00-043-1733, AM show peaks associated to the crystalline planes (011), (021), (111), (102), (112), (121), (032), (130), (131), (104), (124), (105), (220) and (116). The diffractogram indicates that AM shows a trihydrate structure [12].

Figure 2 highlighted that the PLE and PL spectra of AM, which are characterized by bands, peaked at 374 nm and 445 nm, respectively. The UV exposure of AM induces an intensity decrease in the case of: (i) the PLE spectrum from 1.12 × 107 counts/s to 3.43 × 106 counts/s, and (ii) the PL spectrum from 2.16 × 105 counts/s to 1.54 × 105 counts/s. Similar behavior occurs for the AMP drug. Thus, in Figure 3 one observes that: (i) the PLE spectra of AMP show an intensity decrease in the band at 370 nm from 8.46 × 106 counts/s to 2.56 × 106 counts/s occurs, and (ii) the PL spectra of AMP show the intensity decrease in the band at 443 nm from 2.36 × 106 counts/s to 1.04 × 106 counts/s. According to these results, the decrease in intensity of the PL spectra of AM and AMP is of 1.4 and 2.26 times, respectively. This fact indicates that the AM PD is amplified by the presence of the magnesium stearate (MS) excipient, a compound that is added to AM powder in the case of AMP drugs in a solid state. While the profile of the PL spectra of AM and AMP are similar, in the case of PLE spectra of AMP, an additional band at 426 nm (Figure 3a) is observed. Figure 4 demonstrates that this band belongs to MS, the PLE spectrum MS, recorded at the emission wavelength of 475 nm, highlighting a band at 426 nm.

The changes in the intensity of PL and PLE reported in Figure 2 and Figure 3 indicate that a PD of AM and AMP occurs by the UV exposure of these samples. In order to explain the variations reported in Figure 2, Figure 5 shows IR spectra of AM before and after the UV exposure. The IR spectrum of AM shows eleven bands with the maxima at 642, 729, 1122, 1213, 1308, 1379–1392, 1525–1581, 1695, 1774, 2941 and 3335 cm^−1^, they are attributed to the vibrations of deformation in plane of phenyl + torsion HNCCl in latame/amide + stretching Cl in aliphatic group; torsion CNCO in lactam + deformation COO^−^ + stretching SCl form II; deformation COO^−^ + stretching C–C in aliphatic + deformation out-of-plane in phenyl; deformation C–H–C(CH_3_) + stretching C–C(CH_3_) + stretching C–N in lactam/amide + deformation H–C–N in lactam; streching C–C + deformation H–N–C(NH_3_^+^) + deformation in-plane C–H in phenyl + deformation H–C–N in lactam; symmetrical stretching COO^−^ + stretching C–Cl in phenyl + deformation in plan C–H in phenyl + stretching CN in amide/lactam + deformation H–C–C(CH_3_) + deformation H–N–C in amide + deformation NH_3_^+^; deformation CH_2_; deformation C–H in phenyl + amide II + deformation NH_3_^+^ + asymetrical stretching COO^−^; amide I + deformation N–H; stretching C=O in lactamic ring; stretching C–H in aliphatic; and stretching N–H, respectively [12].

After the UV exposure, the following changes are observed in the IR spectrum of AM: (i) the emergence of a new IR band with a maximum at 3445 cm^−1^; (ii) the change in the absorbance ratio of the IR bands peaked at 1774, 1693–1695, 1525 and 1308 cm^−1^ (A_1774_/A_1693–1695_, A_1774_/A_1525_, A_1774_/A_1308_) from 1.18, 1.39 and 1.42 (Figure 5a) to 1.31, 1.53 and 1.56 (Figure 5b); and (iii) the absorbance ratio of the IR bands at 1774 and 2941 cm^−1^ is changed from 6.36 (Figure 5a) to 6.75 (Figure 5b). The IR band at 3445 cm^−1^ was often associated with the vibrational modes of N–H and O–H groups [19]. The IR band at 3445 cm^−1^ can be explained considering a similar mechanism with that published in Ref. [20], which involves the opening of the β-lactam ring resulting in the emergence of new –COOH and N–H groups. In our case, such a reaction occurs as a result of the reaction of AM with the water vapors from the air.

### 2.2. The Photodegradation of AM and AMP in the Presence of the Alkaline Media

Figure 6 shows the PLE and PL spectra of AM and AMP and their behavior under UV light. 

The PLE spectra of AM and AMP show a band at 351–352 nm with an intensity equal to 8.55 × 10^6^ counts/s and 1.06 × 10^7^ counts/s, respectively, while the PL spectra of AM and AMP are characterized by an emission band at 442 nm and 439 nm, and the intensity is equal to 2.47 × 10^4^ counts/s and 9.49 × 10^5^ counts/s (Figure 6). The UV exposure of the AM and AMP aqueous solutions, time of 273 min, induces as main changes (Figure 6): (i) an intensity decrease in the PLE spectra up to 7.42 × 10^6^ counts/s and 9.74 × 10^6^ counts/s while (ii) the PL spectra intensity is changed to 1.51 × 10^4^ counts/s and 8.2 × 10^5^ counts/s. Knowing these changes, Figure 7 and Figure 8 highlight the changes induced to the PL and PLE spectra by the interaction of AM and AMP, respectively, with NaOH as well as the evolution of PLE and PL spectra when the samples are UV irradiated.

Careful analysis of Figure 7 highlights that when increasing the NaOH weight in the mass of the mixture of AMP and NaOH, one observes that:(i)The PLE spectra show a band at 363 nm (Figure 7a_1_), 361 nm (Figure 7b_1_) and 348 nm (Figure 7c_1_), its intensity is equal to 3.7 × 10^7^ counts/s (Figure 7a_1_), 4.46 × 10^7^ (Figure 7b_1_) and 6.85 × 10^7^ (Figure 7c_1_); the UV exposure of the samples induces a shift of the PLE band to 375 nm (Figure 7a_1_), 352 nm (Figure 7b_1_) and 356 nm (Figure 7c_1_); the intensity is equal to 3.44 × 10^7^ counts/s (Figure 7a_1_), 3.17 × 10^7^ counts/s (Figure 7b_1_) and 5.09 × 10^7^ counts/s (Figure 7c_1_);(ii)The PL band is peaked at 431 nm (Figure 7a_2_), 430 nm (Figure 7b_2_) and 439 nm (Figure 7c_2_) with an intensity equal to 2 × 10^6^ counts/s (Figure 7a_2_), 1.6 × 10^6^ counts/s (Figure 7b_2_) and 7.69 × 10^5^ counts/s (Figure 7c_2_); after the UV exposure of the three samples, the PL spectra intensity is equal to 1.64 × 10^6^ counts/s (Figure 7a_2_), 1.11 × 10^6^ counts/s (Figure 7b_2_) and 2.12 × 10^5^ counts/s (Figure 7c_2_).

Figure 8 highlights the PLE and PL spectra of NaOH reacted AM, when the NaOH weight increases more and more, the following changes: (i) the PLE spectra, before to UV exposure, show a maximum of PLE band peaked at 371 nm (Figure 8a_1_), 362 nm (Figure 8b_1_) and 360 nm (Figure 8c_1_), the intensity is equal to 1.08 × 10^7^ counts/s (Figure 8a_1_), 1.65 × 10^7^ counts/s (Figure 8b_1_) and 2.5 × 10^7^ counts/s (Figure 8c_1_); after the UV exposure, the PLE bands are localized at 373 nm (Figure 8a_1_), 362 nm (Figure 8b_1_) and 353 nm (Figure 8c_1_) with the intensity equal to 2.17 × 10^7^ counts/s (Figure 8a_1_), 2.95 × 10^7^ counts/s (Figure 8b_1_) and 3.67 × 10^7^ counts/s; (ii) the PL spectra show an emission band peaked at 430 nm (Figure 8a_2_), 444 nm (Figure 8b_2_) and 450 nm (Figure 8c_2_) with the intensity equal to 1 × 10^6^ counts/s (Figure 8a_2_), 6.85 × 10^5^ counts/s (Figure 8b_2_) and 6.02 × 10^3^ counts/s (Figure 8c_2_). The UV exposure of the samples induce an intensity increase in the PL band up to 1.71 × 10^6^ counts/s (Figure 8a_2_), 1.09 × 10^6^ counts/s (Figure 8b_2_) and 2.15 × 10^4^ counts/s (Figure 8c_2_).

These changes are induced by the emergence of a hydrolysis compound under UV light. In order to sustain this sentence, Figure 9 shows the UV-VIS spectra of AM reacted with NaOH and the evolution of these spectra when the samples are UV exposed.

As observed in Figure 9a, the UV-VIS spectra of AM reacting with NaOH show a band of high absorbance at 258 nm and another one of low absorbance peaked at 324 nm; these are assigned to electronic transitions n-π* and π-π* [21]. These two bands are localized at 264 nm and 328 nm in the case of the UV-VIS spectrum of NaOH-reacted AMP (Figure 9b). The UV exposure of these samples induces a gradual absorbance increase in the band in the 300–375 nm range simultaneous with the absorbance decrease in the band situated in the 240–300 nm range, variation, which is accompanied by the emergence of an isosbestic point at ~300 nm. This indicates the generation of a new compound when the samples AM: NaOH and AMP: NaOH are exposed to UV light.

In order to explain the chemical processes assisted of UV light, Figure 10 and Figure 11 show Raman and IR spectra of the AM: NaOH sample. The Raman spectrum of AM (Figure 10a) shows seventeen lines at la cca. 590, 781, 833, 953, 1005, 1157, 1188, 1236, 1321, 1460, 1605, 1695, 1766, 2941, 2987, 3064 and 3336 cm^−1^; these are associated to the vibrational modes of out-of-plane deformation of benzene ring, deformation in the plane of phenyl + stretching C–C aliphatic + deformation of the C–N–C bond in lactam structure; deformation of N–C–O in lactam + rotation C–N–C–O in lactam + deformation in plan of phenyl; stretching C–C in aliphatic + deformation in plan O–C–O–C+ deformation C–C–C in lactam chain + deformation out-of-plan C–H in phenyl; deformation C–N–C in amide, lactame + stretching C–C in lactam+ deformation out-of-plan C–H in phenyl + deformation H–C–C–H(CH_3_); deformation H–C–C in phenyl + deformation out of plan C–H in phenyl + stretching C–C in phenyl + deformation H–C–C–H(CH_3_) + deformation NH_3_^+^; deformation in plan H–C–C in phenyl + deformation C–H + stretching C–C; deformation H–N–C in lactam/amide + stretching C–N in amide + deformation H–C–C + deformation NH_3_^+^; stretching C–C + deformation H–N–C(NH_3_^+^) + deformation H–C–N in lactam; deformation in plan C–H in phenyl + deformation H–C–C(CH_3_) + stretching C–N in amide/lactam; stretching C–Cl in phenyl + deformation C–H in phenyl + deformation CH_3_/CH_2_; stretching C–Cl in phenyl + deformation C–H in phenyl +deformation NH_3_^+^; amide I + deformation N–H; stretching C=O in lactamic ring; stretching C–H in aliphatic; stretching C–H in aromatic ring and stretching N–H [12]. 

Unlike Figure 10a, the Raman spectrum of the AM: NaOH sample (Figure 10b) highlights that: (i) the most intense Raman line is peaked at 1004 cm^−1^; (ii) the emergence of new Raman lines peaked at 1080, 1394 and 1630 cm^−1^; (iii) a down-shift of the Raman line from 1188 cm^−1^ to 1184 cm^−1^; (iv) a significand intensity decrease in the Raman line at 2939 cm^−1^ so that the intensities ratio of the Raman lines from 2941 to 2939 and 3064 to 3062 cm^−1^ (I_2941–2939_/I_3064–3062_) varies from 16.16 (Figure 10a) to 0.67 (Figure 10b); and (v) the disappearance of the Raman line at 781 cm^−1^. These variations can be explained if we accept that photochemical interaction of AM with NaOH takes place according to Figure 1.

The interaction of the organic compounds containing the amide groups with an alkaline medium such as NaOH, known under the name of the hydrolysis reaction, was demonstrated to lead to the generation of new compounds containing the type primary amines and sodium salts of carboxylic acid [22,23]. In this context, the first two compounds in Figure 1 correspond to 2-amino-2-phenyl acetate sodium (compound 1) and 2-amino-3,3-dimethyl-7-oxo-4-thia-1-azabicyclo[3.2.0]heptane-2-carboxylic acid (compound 2). The generation of α-aminobenzylpenicilloic acid sodium salt (compound 3 in Figure 1) was also reported by I.R. Misic et al. [24]. In our opinion, Figure 1 explains the changing I_2943–2939_/I_3066–3062_ ratio and the new Raman lines at 1080, 1394 and 1630 cm^−1^ associated with the vibrational modes –COONa, C–N–C stretching in amine group + C–C–O stretching and C=O stretching + C–N stretching + C–C–N deformation in amide group, respectively [22,23,25]. An additional experimental argument for the increase in the C=O weight in reaction products of Figure 1 is shown in Figure 11.

The interaction of AM with NaOH induces the following variations in Figure 11 by reference to Figure 5b: (i) a decrease in the ratio between the absorbance of the IR bands at 3335 and 1774 cm^−1^, assigned to the vibrational modes of stretching of N–H and C=O bonds in carboxyl groups and lactamic rings [12,25] from 0.608 (Figure 5b) to 0.347 (Figure 11); (ii) an increase in the ratio between the absorbance of the IR bands at 1774 and 1525 cm^−1^, attributed to the vibrational modes of C=O bonds in carboxyl groups and lactamic rings and stretching CH_2_ [12,25] from 1.53 (Figure 5b) to 2.13 (Figure 11). These variations clearly indicate an increase in the weight of the vibrational modes of the C=O bonds in the carboxyl groups and the lactam ring existent in reaction products of Figure 1.

## 3. Materials and Methods

Ampicillin (abbreviate AM) and NaOH were purchased from Sigma-Aldrich. The drug marked under the name Ampicillin (abbreviate AMP) was bought from a local pharmacy. The composition of an AMP tablet was 500 mg AM, magnesium stearate and talc. Both AM and excipients were stocked in two capsules of yellow and red color containing quinoline yellow, TiO_2_, methyl p-hydroxybenzoate, propyl p-hydroxybenzoate and gelatin.

The aqueous solutions of AM and AMP with the concentration of 50 mg/mL and NaOH 0.3 M were prepared and mixed in the volumetric ratio equal to 2:1, 1.5:1.5 and 1:2. The AMP PD process was studied by removing the two capsules, the powder inside them consisting of AM, magnesium stearate (MS) and talc was dispersed/dissolved in water by the ultrasonication and then removing of talc and MS by filtration was achieved. Elimination of MS and talc allows a correct assessment of the influence of NaOH on aqueous solutions of ampicillin prepared from compounds purchased from Sigma-Aldrich (AM) and the local pharmacy (AMP).

The UV-VIS spectra of AM interacted to NaOH were recorded with a UV-VIS-NIR spectrophotometer, Lambda 950 model, from Perkin Elmer, and the scan speed was 266.75 nm/min.

The photoluminescence (PL) and photoluminescence excitation (PLE) spectra of AM and AMP were recorded with an FL3-22Fluorolog spectrometer, from Horiba Jobin Yvon, in right-angle geometry, with a Xe lamp as excitation source with the power of 450 W. The excitation and emission wavelengths used for the recording of the PL and PLE spectra were equal to 330 and 500 nm, respectively, and the integration time was 0.5 s.

Raman spectra of AM were recorded with a Bruker MultiRam FT Raman spectrophotometer, with a YAG:Nd laser as the excitation source (the wavelength excitation was 1064 nm), in backscattering macroscopic geometry, with a resolution of 1 cm^−1^. Other experimental conditions used in this order consist of: (i) a laser power of 50 mW; (ii) a scan number of 200.

IR spectra of AM were recorded with a Vertex 80 FTIR spectrophotometer from Bruker; in the attenuated total reflection geometry, the scan number was equal to 64, and the resolution was 2 cm^−1^.

The samples analyzed by the IR spectroscopy and the Raman scattering were prepared using 1 mL from the solutions of AM and NaOH-reacted AM, which were previously photodegraded; these were deposited onto a Si plate with a length x width of 1 cm^2^ and then dried at 100 °C, under vacuum, for 1 h. Crystallized powders were collected and used in the studies by the Raman scattering and the IR spectroscopy.

Diffractogram of AM was recorded with a D8 Advanced Bruker X-ray diffractometer, using CuKα radiation (λ = 1.54056 Å).

## 4. Conclusions

In this article, new results by photoluminescence, FTIR spectroscopy, Raman scattering and UV-VIS spectroscopy were reported for the PD of AM and AMP induced by NaOH. These results allow us to conclude that: (i) AM used in AMP shows a trihydrate crystalline structure; (ii) the variation in the intensity of the PL and PLE spectra of the AM and AMP has suggested that a PD took place; (iii) the interaction of AM with NaOH induces the emergence of new compounds highlighted by the appearance of an isosbestic point in UV-VIS spectra; (iv) according to Raman scattering, the PD products of the NaOH-reacted AM contain the functional groups of the type -COONa, C–N–C in amine and C=O, which were highlighted by the Raman lines at 1080, 1394 and 1630 cm^−1^.

## Data Availability

Data is contained within the article.

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
