# Peer review of "New Aspects Concerning the Ampicillin Photodegradation"

_pharmaceuticals, 2022, doi:10.3390/ph15040415_

Round 1

Reviewer 1 Report

-section of materials and methods should be modified with more details like laser source of Raman and objective and the time of scan.... ,etc.

-Figure 8 or 9?? "Raman" recommend to remeasure again without baseline to present the bands in good form.

-check the number of Figures as Figure 8 (it is repeated twice time) check it carefully.

-"Unlike Figure 9a, the Raman spectrum of the AM : NaOH sample (Figure 9b) highlights that: i) as most intense Raman line is peaked at 1004 cm-1; ii) the emergence of new Raman lines peaked at 1080, 1394 and 1630 cm-1; iii) a down-shift of the Raman lines from 1190 and 2943 cm-1 to 1184 and 2939 cm-1; iv) a significand intensity decrease of the Raman line at 2939 cm-1 so that the intensities ratio of the Raman lines from 2943-2939 and 3066-3062 cm-1 (I2943-2939/I3066-3062) varies from 1.57 (Figure 9a) to 0.67 (Figure 9b); and v) the disseapearance of the Raman line at 783 cm-1. These variations can be explain if we accept that photochemical interaction of AM with NaOH takes place according to Scheme 1."

where is the reference which recommend this hypothesis ??

-you should update the references with recent studies

Author Response

-section of materials and methods should be modified with more details like laser source of Raman and objective and the time of scan.... ,etc.

Authors reply: In the revised manuscript, at the section of Materials and Methods have been included the following comments: 

“The UV-VIS spectra of AM interacted to NaOH were recorded with a UV-VIS-NIR spectrophotometer, Lambda 950 model, from Perkin Elmer, the scan speed being of 266.75 nm/min.   

The photoluminescence (PL) and photoluminescence excitation (PLE) spectra of AM and AMP were recorded with a FL3-22Fluorolog spectrometer, from Horiba Jobin Yvon, in right-angle geometry, having as excitation source a Xe lamp with the power of 450 W. The excitation and emission wavelengths used for the recording of the PL and PLE spectra, were equal to 330 and 500 nm, respectively, the integration time being of 0.5 sec.               

Raman spectra of AM were recorded with a Burker MultiRam FT Raman spectrophotometer, having a YAG:ND laser as excitation source (the wavelength excitation was 1064 nm), in backscattering macroscopic geometry, with a resolution of 1 cm-1. Other experimental conditions used in this order consist in: i) the laser power was of 50 mW; and ii) the scans number was equal to100.           

IR spectra of AM were recorded with a Vertex 80 FTIR spectrophotometer, from Bruker, in the attenuated total reflection geometry, scans number was equal to 64 and the resolution was 2 cm-1.               

The photodegradation process was carried out using solutions above prepared. In the case of the UV-VIS, Raman and IR spectra, a Hg-vapor lamp with the power of 350 W was used. We have selected the 200–380 nm spectral range, where is known that Hg spectral lines at 253nm is most intense, with an optical filter UG5. In the case of the PL and PLE spectra, the photodegradation process was performed using the Xe lamp with the power of 450 W.               

The samples analyzed by the IR spectroscopy and the Raman scattering were prepared using 1 ml from the solutions of AM and NaOH-reacted AM, which were previously photodegraded, these being deposited onto a Si plate with length x width of 1 cm2 and then dried at 100 °C, under vacuum, for 1 hour. Crystallized powders were collected and used in the studies by the Raman scattering and the IR spectroscopy. “

-Figure 8 or 9?? "Raman" recommend to remeasure again without baseline to present the bands in good form.

Authors reply: In the revised manuscript, we have corrected this mistake, the line 184 has been rewritten as follows: “Figure 9. Raman spectra of AM (a) and NaOH-reacted AM (b). In insert of Figures (a) and (b) are shown the Raman spectra after the baseline correction. “  

-check the number of Figures as Figure 8 (it is repeated twice time) check it carefully.

Authors reply: In the revised manuscript, this mistake has been corrected, and the line 184 has been rewritten as follows: “Figure 9. Raman spectra of AM (a) and NaOH-reacted AM (b). In insert of Figures (a) and (b) are shown the Raman spectra after the baseline correction. “   

-"Unlike Figure 9a, the Raman spectrum of the AM : NaOH sample (Figure 9b) highlights that: i) as most intense Raman line is peaked at 1004 cm-1; ii) the emergence of new Raman lines peaked at 1080, 1394 and 1630 cm-1; iii) a down-shift of the Raman lines from 1190 and 2943 cm-1 to 1184 and 2939 cm-1; iv) a significand intensity decrease of the Raman line at 2939 cm-1 so that the intensities ratio of the Raman lines from 2943-2939 and 3066-3062 cm-1 (I2943-2939/I3066-3062) varies from 1.57 (Figure 9a) to 0.67 (Figure 9b); and v) the disseapearance of the Raman line at 783 cm-1. These variations can be explain if we accept that photochemical interaction of AM with NaOH takes place according to Scheme 1." where is the reference which recommend this hypothesis ??

Authors reply: In the revised manuscript, we have included the following comment:“The interaction of the organic compounds containing the amide groups with alkaline medium like NaOH, known under the name of hydrolysis reaction, was demonstrated to lead at the generation of new compounds containing of the type primary amines and sodium salts of carboxylic acid [19, 20 and so on]. In this context, the first two compounds in Scheme 1 correspond to 2-amino-2-phenyl acetate sodium (compound 1) and 2-amino-3,3-dimethyl-7-oxo-4-thia-1-azabicyclo[3.2.0]heptane-2-carboxylic acid (compound 2). The generation of a-aminobenzylpenicilloic acid (compound 3 in Scheme 1) was reported by I.R. Misic et al. [24]. “

-you should update the references with recent studies

Authors reply: In the revised manuscript, we have included the following references:

[9] Lee, H.; Yang, J.W.; Liao, J.D.; Sitjar, J.; Liu, B.H.; Sivashanmugan, K.; Fu, W.E.; Chen, G.D. Dielectric nanoparticles coated upon silver hollow nanosphere as an integrated design to reinforce SERS detection of trace ampicillin in milk solution, Coatings 2020, 10, 390.

[17] Belhacova, L.; Bibova, H.; Marikova, T.; Kuchar, M.; Zouzelka, R.; Rathousky, J. Removal of ampicillin by heterogeneous photocatalysis: combined experimental and DFT study, Nanomaterials 2021, 11, 1992.

[18] Bobirica, C.; Bobirica, L.; Rapa, M.; Matei, E.; Predescu, A.M.; Orbeci, C. Photocatalytic degradation of ampicillin using PLA/TiO2 hybrid nanofibers coated on different types of fiberglass, Water 2020, 12, 176.

[24] Misic, I.R.; Miletic, G.; Mitic, S.; Mitic, M.; Pecev-Marinkovic, E. A simple method for ampicillin determination in pharmaceuticals and human urine, Chem. Pharm. Bull. 2013, 61, 913-919.

An English language check was performed. Some examples are:

- line 56:  the sentence “The UV expsure to AM induces.... “ has been rewritten as follows : “The UV exposure of AM induces “;

-line 182 – “v) the disseapearance “ has been rewritten “v) the dissapearance “;

- line 288-291: the sentence “iv) according to Raman scattering, the PD products of the NaOH-reacted AM contain the functional groups of the type -COONa, C-N-C in amine and C=O, highligthed by the Raman lines at 1080, 1394 and 1630 cm-1.“ has been rewritten as follows: “iv) according to Raman scattering, the PD products of the NaOH-reacted AM contain the functional groups of the type -COONa, C-N-C in amine and C=O, which were highlighted by the Raman lines at 1080, 1394 and 1630 cm-1. “

Reviewer 2 Report

As a whole, the article is interesting. Bearing in mind the extensive use of ampicillin in practice, the reported research may be of great interest not only from a scientific point of view, but also from a practical point.

However, the article contains some editorial shortcomings. It does not contain any information on how the photodegradation process was carried out. Only wavelength and time of irradiation were given. What was the source or irradiation, other conditions like solid layers, etc. were not mentioned.

It is also unclear what the authors understand as degradation in the presence of excipients. It is not clear to me:

 “The AMP PD process was studied removing the two capsules, powder was dissolved in water by the ultrasonication and then removing of excipients by filtration was achieved.”

Which substances had been dissolved?  Ampicyllin? Which excipients should be taken into account when potential interactions were suggested?

Some parts of Results section sound like Discussion:

“These changes suggest a PD of AM and AMP induced in environmental conditions by the UV exposure of the samples.”

Author Response

However, the article contains some editorial shortcomings. It does not contain any information on how the photodegradation process was carried out. Only wavelength and time of irradiation were given. What was the source or irradiation, other conditions like solid layers, etc. were not mentioned.

Authors reply: In the revised manuscript, we have included the following comments:              

“The UV-VIS spectra of AM interacted to NaOH were recorded with a UV-VIS-NIR spectrophotometer, Lambda 950 model, from Perkin Elmer, the scan speed being of 266.75 nm/min.              

The photoluminescence (PL) and photoluminescence excitation (PLE) spectra of AM and AMP were recorded with a FL3-22Fluorolog spectrometer, from Horiba Jobin Yvon, in right-angle geometry, having as excitation source a Xe lamp with the power of 450 W. The excitation and emission wavelengths used for the recording of the PL and PLE spectra, were equal to 330 and 500 nm, respectively, the integration time being of 0.5 sec.               

Raman spectra of AM were recorded with a Bruker MultiRam FT Raman spectrophotometer, having a YAG:Nd laser as excitation source (the wavelength excitation was 1064 nm), in backscattering macroscopic geometry, with a resolution of 1 cm-1. Other experimental conditions used in this order consist in: i) the laser power was of 50 mW; and ii) the scans number was 100.                

IR spectra of AM were recorded with a Vertex 80 FTIR spectrophotometer, from Bruker, in the attenuated total reflection geometry, scans number was equal to 64 and the resolution was 2 cm-1.              

The photodegradation process was carried out using solutions above prepared. In the case of the UV-VIS, Raman and IR spectra, a Hg-vapor lamp with the power of 350 W was used. We have selected the 200–380 nm spectral range, where is known that Hg spectral lines at 253nm is most intense, with an optical filter UG5. In the case of the PL and PLE spectra, the photodegradation process was performed using the Xe lamp with the power of 450 W.               

The samples analyzed by the IR spectroscopy and the Raman scattering were prepared using 1 ml from the solutions of AM and NaOH-reacted AM, which were previously photodegraded, these being deposited onto a Si plate with length x width of 1 cm2 and then dried at 100 °C, under vacuum, for 1 hour. Crystallized powders were collected and used in the studies by the Raman scattering and the IR spectroscopy. “

It is also unclear what the authors understand as degradation in the presence of excipients. It is not clear to me:

 “The AMP PD process was studied removing the two capsules, powder was dissolved in water by the ultrasonication and then removing of excipients by filtration was achieved.” Which substances had been dissolved?  Ampicyllin? Which excipients should be taken into account when potential interactions were suggested?

Authors reply: In the revised manuscript, the above sentence has been rewritten as follows:“The AMP PD process was studied removing the two capsules, the powder inside them consisting of AM, magnesium stearate (MS) and talc was dispersed/dissolved in water by the ultrasonication and then removing of talc and MS by filtration was achieved.“  According to the section “2.1. Optical and structural properties of AM and its photodegradation in solid state, the presence MS is observed only in the PLE spectra of the AMP drug in solid state. With the help of Figure 4, we have demonstrated that the 426 nm band observed in the PLE spectra of AMP (Figure 3a) corresponds to the electronic transition of MS.  In the section “2.2. The photodegradation of AM and AMP in the presence of the alkaline media “, the experimental results are not influenced of MS, because the samples in liquid state are prepared as was described in the section Materials and Methods. According to lines 240-245, the preparation of the samples used in section 2.2 involves: “The AMP PD process was studied removing the two capsules, the powder inside them consisting of AM, magnesium stearate (MS) and talc was dispersed/dissolved in water by the ultrasonication and then removing of talc and MS by filtration was achieved. Elimination of MS and talc allows a correct assessment of the influence of NaOH on aqueous solutions of ampicillin prepared from compound purchased from the Aldrich-Sigma company (AM) and local pharmacy (AMP).“

Some parts of Results section sound like Discussion:

“These changes suggest a PD of AM and AMP induced in environmental conditions by the UV exposure of the samples.”

Authors reply: In the revised manuscript, two new figures, i.e. Figures 4 and 11, and the following comments have been included in the Results and Discussion section:

-line 63-78: “According to these results, the decrease of intensity of the PL spectra of AM and AMP is of 1.4 and 2.26 times, respectively. This fact indicates that the AM PD is amplified of the presence of the magnesium stearate (MS) excipient, compound that is added to AM powder in the case of AMP drugs in solid state. While the profile of the PL spectra of AM and AMP are similar, in the case of PLE spectra of AMP an additional band at 426 nm (Figure 3a) is observed. Figure 4a demonstrates that this band belongs to MS, the PLE spectrum MS, recorded at the emission wavelength of 475 nm, highlighting a band at 426 nm.……

Figure 4. PLE spectrum of the MS, recorded at the emission wavelength of 475 nm.

 The changes of the intensity of PL and PLE reported in Figures 2 and 3 indicate that a PD of AM and AMP occurs by the UV exposure of these samples. To explain the variations reported in Figure 2, Figure 5 shows IR spectra of AM before and after the UV exposure. “ 

- line 214-238: “The interaction of the organic compounds containing the amide groups with alkaline medium like NaOH, known under the name of hydrolysis reaction, was demonstrated to lead at the generation of new compounds containing of the type primary amines and sodium salts of carboxylic acid [22, 23 and so on]. In this context, the first two compounds in Scheme 1 correspond to 2-amino-2-phenyl acetate sodium (compound 1) and 2-amino- 3,3-dimethyl-7-oxo-4-thia-1-azabicyclo[3.2.0]heptane-2-carboxylic acid (compound 2). The generation of a-aminobenzylpenicilloic acid salt sodium (compound 3 in Scheme 1) was also reported by I.R. Misic et al. [24]. In our opinion, Scheme 1 explains the changing I2943-2939/I3066-3062 ratio and the new Raman lines at 1080, 1394 and 1630 cm-1 associated to the vibrational modes -COONa, C–N–C stretching in amine group + C–C–O stretching and C=O stretching + C-N stretching + CCN deformation in amide group, respectively [22, 23, 25]. An additional experimental argument for the increase the C=O weight in reaction products of Scheme 1 is shown in Figure 11. …..Figure 11. The IR spectrum of NaOH-reacted AM The interaction of AM with NaOH induces the following variations in Figure 11 by reference to Figure 5b: i) a decrease of the ratio between the absorbance of the IR bands at 3335 and 1774 cm-1, assigned to the vibrational modes of stretching of NH and C=O bonds in carboxyl groups and lactamic rings [12, 25] from 0.608 (Figure 5b) to 0.347 (Figure 11); and ii) an increase of the ratio between the absorbance of the IR bands at 1774 and 1525 cm-1, attributed to the vibrational modes of C=O bonds in carboxyl groups and lactamic rings and stretching CH2[12, 25] from 1.53 (Figure 5b) to 2.13 (Figure 11). These variations clearly indicate an increase in the weight of the vibrational modes of the C = O bonds in the carboxyl groups and the lactam ring existent in reaction products of Scheme 1.“

An English language check was performed. Some examples are:

- line 56:  the sentence “The UV expsure to AM induces.... “ has been rewritten as follows : “The UV exposure of AM induces.... “;

-line 182 – “v) the disseapearance “ has been rewritten “v) the dissapearance “;

- line 288-291: the sentence “iv) according to Raman scattering, the PD products of the NaOH-reacted AM contain the functional groups of the type -COONa, C-N-C in amine and C=O, highligthed by the Raman lines at 1080, 1394 and 1630 cm-1.“ has been rewritten as follows: “iv) according to Raman scattering, the PD products of the NaOH-reacted AM contain the functional groups of the type -COONa, C-N-C in amine and C=O, which were highlighted by the Raman lines at 1080, 1394 and 1630 cm-1. “

Round 2

Reviewer 1 Report

Update the references with recent studies as DOI: 10.1016/j.apmt.2021.101129

Reviewer 2 Report

The manuscript was extensively corrected. All my suggestions were taken into account. Now, the methods are clearly described.